# Change in Diagnosis of *Helicobacter pylori* Infection in the Treatment-Failure Era

**DOI:** 10.3390/antibiotics13040357

**Published:** 2024-04-12

**Authors:** Rocco Spagnuolo, Giuseppe Guido Maria Scarlata, Maria Rosaria Paravati, Ludovico Abenavoli, Francesco Luzza

**Affiliations:** Department of Health Sciences, University “Magna Graecia”, 88100 Catanzaro, Italy; spagnuolo@unicz.it (R.S.); giuseppeguidomaria.scarlata@unicz.it (G.G.M.S.); mrparavati@unicz.it (M.R.P.); l.abenavoli@unicz.it (L.A.)

**Keywords:** antimicrobial resistance, multidrug-resistant, diagnostic approaches, whole genome sequencing

## Abstract

*Helicobacter pylori* (*H. pylori*) infection is a prevalent global health issue, associated with several gastrointestinal disorders, including gastritis, peptic ulcers, and gastric cancer. The landscape of *H. pylori* treatment has evolved over the years, with increasing challenges due to antibiotic resistance and treatment failure. Traditional diagnostic methods, such as the urea breath test, stool antigen test, and endoscopy with biopsy, are commonly used in clinical practice. However, the emergence of antibiotic-resistant strains has led to a decline in treatment efficacy, necessitating a re-evaluation of common diagnostic tools. This narrative review aims to explore the possible changes in the diagnostic approach of *H. pylori* infection in the era of treatment failure. Molecular techniques, including polymerase chain reaction and whole genome sequencing, which have high sensitivity and specificity, allow the detection of genes associated with antibiotic resistance. On the other hand, culture isolation and a phenotypic antibiogram could be used in the diagnostic routine, although *H. pylori* is a fastidious bacterium. However, new molecular approaches are promising tools for detecting the pathogen and its resistance genes. In this regard, more real-life studies are needed to reveal new diagnostic tools suitable for identifying multidrug-resistant *H. pylori* strains and for outlining proper treatment.

## 1. Introduction

*Helicobacter pylori* (*H. pylori*) is a Gram-negative, spiral-shaped, microaerophilic bacterium that can colonize the human gastric mucosa. For this reason, it is defined as a “true resident” of the gastric microbiota, which also contains other bacteria from the gut and oral cavity [1]. The global prevalence of *H. pylori* infection is decreasing: from 58% in the 1980s–1990s, to 43% in recent years. However, significant variations have been observed across different geographic regions. Latin America and the Caribbean exhibited the highest prevalence at 59%, whereas North America had the lowest at 26%. Nationally, Nigeria recorded the highest prevalence at 90%, while Yemen showed the lowest among children aged 10 years at 9%. Disease prevalence showed disparity based on development status, with 51% in developing countries compared to 35% in developed countries, but remained consistent across genders [2,3]. It is responsible for gastritis, peptic ulcers, and gastric cancer, although individuals colonized by the microorganism are often asymptomatic and just develop nausea, vomiting, abdominal pain, and dyspepsia [4]. These pathological expressions arise from the interaction between the bacterium and the host, facilitated by specific virulence factors. *H. pylori* produces the enzyme urease, which catalyzes the hydrolysis of urea into carbon dioxide (CO_2_) and ammonia, counteracting the acidic pH of the stomach. Host colonization is promoted by flagella and adhesins that allow adhesion and mobility of the bacterium in its microenvironment. At the same time, cytotoxin A-associated gene (*cagA*) and vacuolating cytotoxin A (*vacA*) induce an inflammatory pathway by stimulating the production of eosinophils, neutrophils, and mast and dendritic cells. As a result, the gastric epithelial layer also secretes chemokines to initiate innate immunity and activates neutrophils that further damage the host tissue, leading to gastritis and ulcer formation [5,6]. Currently, *H. pylori* is responsible for more than 60% of gastric cancers. Indeed, epidemiological studies show that 2–3% of *H. pylori*-infected people develop gastric adenocarcinoma, and 0.1% will develop mucosa-associated lymphoid tissue lymphoma [7]. *H. pylori* infection can affect the onset of other diseases. The changes in gastric pH induced by *H. pylori* cause changes in the composition of gastric microbiota. The presence of new commensal bacteria in the stomach may affect the inflammatory response already activated by *H. pylori* [8]. Several studies have shown a relationship between *H. pylori* infection and Non-Alcoholic Fatty Liver Disease (NAFLD). In particular, *H. pylori* infection appears to predispose patients to insulin resistance, which can then lead to NAFLD [9,10,11]. Rapid and accurate diagnosis is an important step in patient care, as it allows intervention in the early stages of *H. pylori* infection. Subsequent eradication of the bacterium makes it possible to restore the physiological condition of the gastric mucosa. Traditional diagnostic methods, such as urea breath tests, stool antigen tests, and endoscopy with biopsy, are highly sensitive and inexpensive [12]. Eradication therapy typically consists of a combination of antibiotics and acid-suppressing medications, aimed at effectively eliminating the bacterium and reducing the risk of associated gastrointestinal diseases. According to the Maastricht IV/Florence Consensus Report, commonly recommended first-line therapies include triple therapy, which combines a proton pump inhibitor (PPI) with two antibiotics such as clarithromycin and amoxicillin or metronidazole, administered for a duration of 7 to 14 days. In cases where clarithromycin resistance is prevalent or suspected, alternative first-line regimens may be recommended, such as bismuth quadruple therapy, which includes a PPI, bismuth, tetracycline, and metronidazole or clarithromycin. Sequential therapy, which involves administering a PPI and amoxicillin for the first 5 days followed by a PPI, clarithromycin, and tinidazole or metronidazole for the next 5 days, is another option. For patients who fail to respond to first-line therapy or who have persistent infection after treatment, second-line or rescue therapies may be necessary. These may include different treatments such as levofloxacin, rifabutin, or furazolidone, guided by antimicrobial susceptibility testing when available [13]. However, the efficacy of these treatments can be compromised by the emergence of antimicrobial resistant (AMR) strains. According to epidemiological data, resistance rates vary globally, with certain regions experiencing higher rates than others. For instance, in some parts of Asia, resistance to clarithromycin can exceed 50%, while in Europe it is around 15–20% [14]. Furthermore, a recent systematic analysis has shown that 4.95 million deaths were associated with AMR in 2019, with 1.27 million deaths directly attributable to bacterial AMR. Regionally, the overall death rate attributed to resistance was highest in western sub-Saharan Africa, at 27.3 deaths per 100,000, and lowest in Australasia, at 6.5 deaths per 100,000. In this way, the majority part of deaths associated with AMR were attributable to ESKAPE pathogens (vancomycin-resistant *Enterococcus faecium*, methicillin-resistant *Staphylococcus aureus*, *Klebsiella pneumoniae*, *Acinetobacter baumannii*, *Pseudomonas aeruginosa*, and *Enterobacter* spp.) [15]. At the same time, the circulation of multidrug-resistant (MDR) strains, so defined as they are resistant to at least three different classes of antibiotics, is a public health problem [16]. Indeed, the global prevalence of MDR *H. pylori* is quite heterogeneous, ranging from 15% to 50%. Specifically, it ranges from less than 10% in Europe to 20% and 40% in India and Peru, respectively [17]. This ever-growing phenomenon has made it necessary to increase active surveillance policies and change the diagnostic and therapeutic algorithm in clinical practice [18,19]. Several factors contribute to the development of AMR in *H. pylori*. One key factor is the overuse and misuse of antibiotics in clinical practice, which can lead to the selection of resistant strains. Additionally, the microorganism’s ability to adapt and evolve rapidly contributes to the development of AMR. The bacterium can acquire resistance through mutations in its DNA or by acquiring resistance genes from other bacteria through horizontal gene transfer [20]. Furthermore, inadequate treatment regimens, such as insufficient duration or incorrect dosing of antibiotics, can also promote the emergence of resistant strains. Poor patient compliance with treatment regimens is another contributing factor, as incomplete eradication of the infection can select for resistant bacteria. For this reason, the World Health Organization listed the clarithromycin-resistant *H. pylori* as a high-priority pathogen that requires particular attention for its treatment [14]. This narrative review aims to explore the possible changes in the diagnostic approach of *H. pylori* infection in the treatment-failure era.

## 2. Molecular Mechanisms of Antimicrobial Resistance

AMR, along with the reduced number of effective antibiotics against *H. pylori*, constitutes one of the causes of the failure of bacterial eradication therapy [21]. Over the past two decades, this phenomenon has significantly increased worldwide. The rise in AMR rates has been correlated not only with an individual’s prior use of a specific antibiotic or others belonging to the same class of drugs, but also with the widespread consumption of antibiotics in a population [22]. Antibiotic use exerts selective pressure on bacterial populations. This event causes bacterial species to respond to adverse conditions through the establishment of genome mutations that give rise to AMR. In this regard, a recent study analyzed gastric biopsies from multiple stomach regions of 16 *H. pylori*-infected adults and analyzed the genome of 10 *H. pylori* isolates from each biopsy. As reported by the authors, antibiotics can induce severe population bottlenecks and probably play a role in shaping the population structure of *H. pylori* [23]. Understanding the mechanisms leading to the emergence and spread of antibiotic resistance is one of the strategies that can contribute to effective therapies against *H. pylori*. AMR primarily occurs due to genetic changes in the bacterial cell. Other phenomena can contribute to antimicrobial resistance in bacterial cells, including physiological changes (such as overexpression of efflux pumps) and cellular adaptive properties (such as biofilm formation). AMR observed in *H. pylori* seems to be attributed to gene mutations that alter the drug target or inhibit its activation. These mutations are chromosomal and point mutations, particularly missense, nonsense, frameshift, insertion, or deletion mutations. Extra-chromosomal mutations or those leading to the loss or acquisition of a gene are very rarely encountered. Several studies on *H. pylori* have reported three different drug-resistance models: single drug resistance (SDR), MDR, and hetero-resistance (HR). These three models do not mutually exclude each other; instead, they often overlap and are correlated in their molecular mechanisms and clinical implications [21]. Figure 1 summarizes the main SDR mechanisms observed in *H. pylori.*


### 2.1. Single Drug Resistance

Amoxicillin is a β-lactam antibiotic that belongs to the aminopenicillin subclass [24]. High-dose therapy with amoxicillin and PPIs is an effective first-line strategy in the eradication of *H. pylori*, achieving an 89.3% eradication rate [25]. Amoxicillin, by binding penicillin-binding proteins (PBPs), stops their trans-peptidase activity and, consequently, the synthesis of peptidoglycan. As a result, the cell wall of the bacterium will be less stable and less robust [16,21]. Although *H. pylori* eradication rates with amoxicillin-based regimens are very high, the rate of amoxicillin resistance has gradually increased due to the widespread use of amoxicillin in the treatment of various infections. In particular, resistant genotypes have been observed more after failures in *H. pylori* eradication therapy [24,25]. Although *H. pylori* can produce β-lactamase-like proteins, the main resistance mechanism is the reduction in the affinity of amoxicillin at the binding site on PBP1A, PBP2, and PBP3 [16,26]. Mutations on the gene encoding for PBP1A are the most relevant as they alter the structural amoxicillin-binding motifs (SXXK, SXN, and KTG) and the carboxy-terminal sequence. Mutated PBP1A has a low affinity for amoxicillin, allowing *H. pylori* to survive despite high concentrations of antimicrobials in the cell [25]. Levofloxacin is a third-generation fluoroquinolone characterized by bactericidal activity [16,27]. Its bactericidal activity is determined by inhibition of chromosome replication, inhibiting two essential type II topoisomerases: DNA gyrase and topoisomerase IV. These two enzymes modulate chromosome supercoiling required for DNA synthesis, transcription, and cell division [28,29]. The gene material of *H. pylori* only has genes for DNA gyrase, so levofloxacin exerts its antimicrobial activity exclusively on this target [21]. Although levofloxacin has a good eradication rate (approximately 91.5% on susceptible strains), its use falls under second-line regimes due to the occurrence of cross-resistance with other fluoroquinolones, used in urinary and respiratory infections, resulting in increased resistance to levofloxacin [16,27,28]. In *H. pylori*, resistance to levofloxacin is mediated by target-mediated mechanisms on DNA gyrase. In particular, single- or double-point mutations have been observed on gene codings of DNA gyrase, *gyrA* and *gyrB*, respectively [21]. The most relevant mutations involve the Asn87 and Asp91 positions of the *gyrA* gene, located in a region of *gyrA* known as the quinolone resistance determination region (QRDR) [29,30]. Other mutations outside the QRDR have been observed on both *gyrA* and *gyrB*, together with mutations at positions 87 and 91 of gyrB, but have not yet been associated with the appearance of levofloxacin resistance [21]. Clarithromycin is a macrolide class component and is characterized by a minimal inhibitory concentration (MIC) against *H. pylori* [21,31]. The bacteriostatic action of clarithromycin consists of inhibition of protein synthesis through reversible binding to the V-domain loop on the 23S ribosomal RNA gene (rRNA 23S) of the 50S subunit, known as peptidyl transferase, of bacterial ribosomes [21,31]. The spread of clarithromycin resistance can mainly be traced back to the widespread use of macrolides in the treatment of sexually transmitted infections and respiratory tract infections, including COVID-19 [16,32]. In the case of *H. pylori*, resistance to clarithromycin is mainly determined by point mutations on the peptidyl transferase loop of domain V on 23S rRNA. Two copies of the 23S rRNA operon are present in the *H. pylori* genome, and mutations occur on both copies resulting in a highly resistant strain (MIC > 64 mg L^−1^). However, the heterozygous phenotype can also manifest resistance, resulting in an intermediate-resistance-level strain (0.5 mg ≤ MIC ≤ 1 mg). These mutations alter the structure of the loop, reducing its affinity for the drugs. A total of 90% of resistance-inducing mutations mainly affect the adjacent nucleotide positions 2124 and 2143, where adenine is replaced by guanine in both, and less frequently by a cytosine in 2142 [30]. A second mechanism of clarithromycin resistance in *H. pylori* is the multidrug efflux systems, in particular the resistance-nodulation-cell division family of efflux pumps. A synergistic effect in clarithromycin resistance has been proposed between the mutations and efflux pumps [30,31]. Metronidazole is a prodrug of the antimicrobial class of nitroimidazoles [21]. To carry out its bactericidal activity, metronidazole requires reductive activation mediated by the oxygen-insensitive nitro-reductase (NADPH), encoded by the *rdxA* genes. The activity of NADPH allows the conversion of metronidazole into cytotoxic metabolites, which cause cell death [21,33]. Wide use of nitroimidazoles for *H. pylori* infections and other pathogens has been accompanied by an increase in metronidazole resistance rates in *H. pylori* [20]. Currently, the resistance rate of *H. pylori* to metronidazole is about 68.4% [30]. In *H. pylori* the main mechanism of resistance to metronidazole is determined by a reduced activation of the drug, due to mutations in the NADPH *rdxA* and *frxA* genes [30]. Mutations have been observed in metronidazole-resistant strains on *fdxB*, *fdxA*, and *fldA* genes, which code for ferredoxin-like protein, ferredoxin, and flavodoxin, respectively [33]. Tetracycline is a bactericidal antibiotic of the tetracycline family [16,21]. This antibiotic binds to the 30S subunit of ribosomes and blocks the binding of aminoacyl-tRNA, resulting in stopping protein synthesis [34]. In the case of *H. pylori*, resistance mechanisms have been poorly studied due to a severe lack of isolated strains; however, mutations on rRNA 16S are the main resistance mechanism [21]. Within the 30S subunit, tetracycline has a primary binding site and several secondary binding sites, establishing many hydrophobic interactions. Tetracycline resistance is determined by single, double, or triple mutations on the base pairs AGA926-928 TTC. This triple base pair is located at the primary tetracycline binding site, influencing the drug-ribosome affinity [35]. Generally, this mutation occurs in both pairs of rRNA 16S [34]. Rifabutin is an antibiotic derived from rifamycin and is a great drug for the eradication of *H. pylori* [36]. Its bactericidal activity is determined by binding to the β subunit of DNA-dependent RNA-polymerase, encoded by the *rpoB* gene. Following the binding of rifabutin to DNA-dependent RNA polymerase, RNA synthesis is stopped in the early stages of transcription. Rifabutin has found widespread use in rescue therapy after initial unsuccessful attempts to eradicate *H. pylori* [21]. Cases of antibiotic resistance have also been reported in *H. pylori*. Most rifabutin-resistant *H. pylori* strains have been observed following the failure of eradication treatment [36]. Rifabutin-resistant *H. pylori* strains are characterized by the appearance of point mutations on the *rpoB* gene, in codons 149, 524–545, 585–586, and 701 [21,37]. The difficulty in eradicating *H. pylori* has led to the search for new therapies against this pathogen. Several studies have investigated the efficacy of aminoglycosides on *H. pylori*. A study by Lee et al. reported that gentamicin and netilmicin have low MICs against *H. pylori* in vitro, opening the possibility for new therapeutic strategies [38]. Currently, no clinical studies confirm the therapeutic efficacy of aminoglycosides for eradication nor studies identifying specific resistance mechanisms on *H. pylori*.

### 2.2. Multidrug Resistance

In addition to the difficulties of the SDR of individual antibiotics, the eradication therapy of *H. pylori* is made even more complex due to the appearance of MDR strains. MDR is a serious worldwide problem due to the excessive use of antibiotics. These primary MDR strains account for about 40% of infections in different parts of the world. All this leads to a sharp decline in the eradication of *H. pylori* [21,39]. MDR in *H. pylori* seems to derive from several mechanisms. The first of these comprises gene mutations. The various gene mutations observed in the SDRs of the drugs used for eradication are manifested in a single strain, resulting in an accumulative MDR profile [40]. A second mechanism is a reduced concentration of antibiotics in the bacterial cell that can be determined either by upregulation of the efflux systems, which export different compounds out of the cell, or by downregulation of membrane proteins or lipopolysaccharides that reduce the absorption of drugs [21]. Although in the stomach *H. pylori* is present more in the planktonic form, some strains can form biofilms in the gastric mucosa. Biofilm allows bacteria to create an environment conducive to their survival, where they can replicate and facilitate the evolution and spread of antibiotic resistance [41]. In addition, *H. pylori* can assume a quiescent state by transforming into coccoid forms to survive stress conditions. This form requires high MIC to be eliminated, resulting in a potential increase in antibiotic resistance due to subsequent ultrastructural membrane changes and metabolic pathways that reduce the effectiveness of the drug [42]. The mechanisms that determine tolerance in coccoid forms must be further investigated. Figure 2 summarizes all the MDR mechanisms of *H. pylori.*

### 2.3. Hetero-Resistance

Hetero-resistance is a phenomenon that consists of the coexistence of one or more subpopulations within a bacterial population, with different levels of resistance to antibiotics. This phenomenon is very common in *H. pylori*, although the mechanisms behind hetero-resistance are not yet clear [43]. In this regard, three hypotheses have been proposed for the development of hetero-resistance: (1) simultaneous infection with different strains of *H. pylori*; (2) evolution of a sensitive strain of *H. pylori* in a drug-resistant variant following antibiotic pressure; and (3) increase in the hetero-resistant population from a susceptible clone due to spontaneous mutations, regardless of exposure to the antibiotic. Hetero-resistance can be considered a step in the bacterium’s evolutionary process towards total antibiotic resistance [21]. A great contribution to this evolutionary process is provided by the anatomical and physiological differences between the antral and ossific gastric mucosa. Thus, an evolutionary push is determined that manifests itself as intragastric migration of bacteria from the same clone and a rapid adaptation to the microinches inside the host; consequently, a bacterial population will consist of several evolutionary subgroups [44].

## 3. Conventional Diagnostic Approaches

Currently, several diagnostic tests are divided into non-invasive and invasive methods, each characterized by its advantages, disadvantages, and limitations. The choice of one test over the other takes into account several factors, such as accessibility of the test, laboratory equipment, and the clinical condition of the patient. Non-invasive methods include respiratory, serological, and stool antigen tests. The principal invasive method is the endoscopy with biopsy [45]. 

### 3.1. Non-Invasive Methods

#### 3.1.1. Urea Breath Test

The urea breath test (UBT) is a non-invasive method for diagnosing *H. pylori* infection and evaluating the therapy’s success [13]. This method is based on the urease activity of *H. pylori*, specifically measuring the difference in the proportion of ^13^C or ^14^C in the exhaled air by mass spectrometry, before and after ingestion of radioactively labeled urea with the carbon isotope 13 or 14 [46]. The labeled urea is converted by ureases into carbon dioxide, which is also labeled. The latter is collected and analyzed during the UBT. The presence of infection will be determined by the presence of ^13^C- or ^14^C-labeled CO_2_ in the breath sample taken 30 min after administering radioactive urea. The absence of results will be interpreted as a negative test result. This method is useful for diagnosis in both adults, and children between 3 and 11 years of age, to whom a reduced amount of marker will be administered [45]. The UBT is characterized by a sensitivity of 96–100% and a specificity of 93–100%. In addition to its non-invasive nature, UBT offers the advantage of providing a complete evaluation, which is not compromised by possible sampling errors. The disadvantages of UBT are the influence of drugs used in eradication therapy (antibiotics, PPIs, or bismuth), the need for specialized equipment to measure CO_2_, the handling of radioactive materials, and the high costs involved in the test [47,48]. A recent meta-analysis performed by Lemos et al. showed that ^13^C-UBT has better diagnostic accuracy than ^14^C-UBT [49]. Specifically, ^13^C-UBT reported sensitivity and specificity values of 96.60% and 96.93%, respectively, compared with 96.15% and 89.84% observed with ^14^C-UBT. The main parameters that determine the effectiveness of the test are the dose of radioactive urea administered, the timing of evaluation, and the type of technique used. Regarding the urea dose, the efficacy of the test was analyzed at different doses of urea. The use of ^13^C-UBT is the non-invasive method of choice for the diagnosis of active infection for the follow-up of *H. pylori* eradication in patients who do not require biopsy. It is also preferred in the case of children and pregnant women, and can detect low levels of *H. pylori* infection [50]. A pilot study performed by Alzoubi et al. compared the diagnostic accuracy between ^13^C-UBT and the fecal antigen test [51]. This study found that ^13^C-UBT had better sensitivity and accuracy than the fecal antigen test. Specifically, ^13^C-UBT reported an accuracy of 86.7% and sensitivity of 94.1%, compared with 76.7% and 76.5% observed for the fecal antigen test. Subsequently, the accuracy of these tests in assessing the success of eradication after six weeks of therapy was observed. Successful eradication was observed in about 77% of patients using the *H. pylori* fecal antigen test, while it was about 67% using ^13^C-UBT. Currently, there are very limited data on the use of UBT post-eradication, as patients involved in studies do not continue follow-up after eradication therapy [52]. It is important to be able to identify a valid non-invasive diagnostic method for *H. pylori* infections so that endoscopy can be used only in limited cases. This is not only because endoscopy is an expensive method, but mainly to increase patient compliance.

#### 3.1.2. Stool Antigen Test

The stool antigen test (SAT) identifies the presence of *H. pylori* by the presence of bacterial antigen in feces, produced by the human body in response to bacterial infection. There are two types of SAT: the enzyme immunoassay (EIA) and rapid immunochromatographic assay (ICA), which use monoclonal or polyclonal antibodies. Generally, EIA results are more reliable than those obtained with ICA, and their accuracy is higher when using monoclonal antibodies than polyclonal antibodies [53]. SAT is used both for the diagnosis of *H. pylori* infection and to assess the outcome of eradication treatment. They are non-invasive, easy-to-handle, low-cost methods with good patient compliance, regardless of age. The test has a good sensitivity of 95.5% and a specificity of 97.6% [45]. Several causes can lead to false negatives, including irregular distribution of antigens in feces, destruction of antigens in constipation, continuous bleeding of the gastrointestinal tract, and low bacterial load in the stomach [46]. SAT should be performed before and after treatment, and as a confirmation method if serological assays are positive [13].

#### 3.1.3. Serological Test

Clinical situations where serological tests can be particularly valuable include bleeding peptic ulcers, gastric cancer, atrophy, and recent antibiotic or PPI use. It is important to note that serology does not indicate an active infection. Immunoglobulin G anti-*H. pylori* decreases gradually after eradication, and a positive test may still be evident months later. Hence, serology is not suitable for confirming eradication. Additional limitations arise from the different strains of *H. pylori*, necessitating the use of locally validated tests. In this way, establishing a well-validated positivity cutoff is crucial. Indeed, a borderline or positive test requires confirmation through UBT or SAT. Despite its high sensitivity and specificity (both 80–95%), while various antigen combinations have been explored to identify markers of gastric cancer evolution, none are currently recommended for practical use [13,45]. Table 1 shows the sensitivity, specificity, and accuracy of each non-invasive method for diagnosing *H. pylori* infection. 

### 3.2. Invasive Test 

#### Endoscopy with Biopsy

Endoscopic procedures are currently considered the gold standard test for assessing the presence of *H. pylori* infection and for providing additional information on abnormalities of the gastric mucosa [13]. This diagnostic method is characterized by high efficiency, even in patients without alarming symptoms or with gastro-esophageal reflux. Endoscopy is combined with biopsy, which then requires histological examination [45]. However, endoscopy has several disadvantages compared to non-invasive tests. Indeed, this method may cause unnecessary injury to the gastric mucosa, involves excessive costs, and is uncomfortable for many patients [54,55]. The progression of *H. pylori* infection is characterized by several very heterogeneous stages, which makes diagnosis very complex with a simple endoscopy. For this reason, new, more sophisticated endoscopic techniques have been introduced [45]. Conventional white light endoscopy (WLI) is the current standard for the evaluation of the mucosa of the gastrointestinal tract due to its accessibility, short endoscopic time, and low cost [56]. Subsequently, image-enhanced endoscopies (IEE) were introduced, such as narrow-band imaging (NBI), linked color imaging (LCI), and blue laser imaging (BLI) [55]. NBI was the first commercial narrow-band technology. The narrow-band illumination is absorbed by hemoglobin and the shortened wavelength penetrates the surface tissue. This technique results in greater contrast of the superficial micro-vessels and the mucosal surface. Narrow-band imaging (M-NBI) is widely used in Asian countries, but not in Western countries [56]. LCI is a color enhancement technology. The output of LCI provides the image with color enhancement in its range, improving mucosal color differences and helping to detect sufficient brightness [57]. Finally, BLI works with two types of lasers with wavelengths of 410 and 450 nm. Its main role is the observation of the target at a short distance, which is called magnification endoscopy [56]. Advanced endoscopic imaging can improve the visualization of the mucosa and vasculature, especially in magnification mode. Many clinical studies have reported that IEE could help to identify the mucosal changes and be used for precisely targeted biopsies. Limitations of using IEE include the need for more training and a learning curve for experience, as well as being time-consuming [55]. Table 2 shows the sensitivity, specificity, and accuracy of each endoscopic technique for diagnosing *H. pylori* infection.

## 4. New Perspectives in Diagnostics and Applicability in Real Life

The diagnostic approaches listed so far are part of common clinical practice. However, the newly highlighted mechanisms of AMR have revealed critical diagnostic and therapeutic issues that need major revisions. The Maastricht IV/Florence Consensus Report recommends that culture isolation and a phenotypic or genotypic antibiogram be routinely performed, even before prescribing first-line treatment, in respect to antibiotic stewardship [13].

### 4.1. Conventional Microbiological Approaches

Conventional microbiological approaches are based on culture isolation and the phenotypic antibiogram. These methods using a phenotypic antibiogram performed after culture isolation from gastric biopsy samples are considered the gold standard for the diagnosis of *H. pylori* infection due to their high specificity (98%), and allow for the definition of a MIC, favoring the use of a tailored therapy [58]. Notably, The Maastricht IV/Florence Consensus Report advises performing antimicrobial susceptibility tests in areas where clarithromycin resistance exceeds 15%. However, these recommendations are difficult to apply in clinical practice. Indeed, the timing of execution of these tests is still much debated, showing a low level of evidence, despite a high level of concordance [13]. Conversely, a recent meta-analysis encompassing 51 distinct studies yielded contrasting findings. In summary, *H. pylori* strains were isolated in 6371 cases (80.7%) out of 7889 infected patients. Culture isolation involved a single antral specimen in 5053 patients, with positive results in 4052 of them (80.1%). When utilizing both antral and gastric body mucosa specimens in 2836 patients, positive results were obtained in 2319 cases (81.7%). Notably, cultures conducted after second- and third-line therapies exhibited a higher success rate compared to those performed before and after the first-line therapy (86.6% vs. 72%, *p* < 0.001). In the broader context, tailored therapies demonstrated a significantly higher success rate compared to empirical treatments (89.7% vs. 77.6%, *p* < 0.001). Eradication rates exhibited significant differences between pre-first-line (91.6% vs. 78.2%; *p* < 0.001), pre-second-line (91.2% vs. 79%, *p* < 0.001), and pre-third-line (79% vs. 70.1%, *p* = 0.03) therapies [59]. Despite the gastric biopsy being the sample of choice, it is also possible to isolate the pathogen from stool samples. In this regard, two pilot studies were performed in the 1990s: Thomas et al. isolated *H. pylori* strain from the stool of 9/23 randomly selected children aged 3–27 months from a Gambian village, while Kelly et al., isolated the pathogen from the stool in 12/25 patients with dyspepsia through conventional culture isolation and polymerase chain reaction (PCR) [60,61]. Another study has shown how, among 50 stool samples, 18 were positive for *H. pylori* [62]. The lack of studies of isolating *H. pylori* from stool samples hinders a precise assessment of the diagnostic impact of this test. While conventional microbiological approaches align with international guidelines, the low sensitivity of culture isolation and the necessity to target specific populations for antibiogram testing discourage its real-world application. In a recent eight-year survey performed in laboratory settings, only 10% use the conventional microbiological approach to isolate *H. pylori* strains [63]. This issue stems from the peculiar characteristics of the pathogen: *H. pylori* is a microorganism defined as “fastidious” because it has specific metabolic characteristics, among which are pH regulation, iron acquisition, and urease production, which make it difficult to isolate in conventional culture media. For this reason, the use of selective culture media and a long incubation period are required (~7 days) [64,65]. Overall, conventional microbiological approaches showed several disadvantages: (i) it is difficult to perform; (ii) it has a long incubation period (~7 days) with high turn-around time (TAT, >7 days); and (iii) it has low sensitivity. These critical issues and low compliance by laboratories in performing this practice make the design of new real-life studies difficult [66].

### 4.2. Molecular Diagnostic Approach

Currently, clarithromycin is the antibiotic of choice in the treatment of *H. pylori* infection. However, the success of pharmacological treatment substantially decreases in cases of antibiotic resistance, rendering this drug ineffective and perpetuating resistance to other bacterial strains. In this regard, the best option would seem to be avoiding therapy with this antibiotic, favoring the initiation of quadruple therapy. Despite the effectiveness of these therapies, adverse events may occur concerning the composition of the gut microbiota and the establishment of new resistance mechanisms. As such, increased assessment of clarithromycin sensitivity through specific diagnostic tests is urgently needed. As previously highlighted, the standard practice in cases of potential resistance in infectious diseases is to conduct an antibiogram. Molecular tests, including PCR, are now commercially available, offering moderate-cost results in a short time. The process involves three key phases: (i) denaturation of the template into single strands; (ii) primer annealing to the strands; and (iii) extension of the new strand [67]. This method not only identifies bacterial DNA using specific *H. pylori* genes (*cagA*, *vacA*, *ureA*, and *ureC*), but also evaluates the presence of genes associated with antibiotic resistance. These include A2143G, A2142G, A2142C (related to clarithromycin); *gyrA* and *gyrB* (linked to levofloxacin); *pbp1A*, *pbp2*, *pbp3*, *hefC*, *hopC*, and *hofH* (related to amoxicillin); and *TET-1* (related to tetracycline) [68]. The diagnostic application of molecular techniques has shown a 91% agreement among expert opinions and a high level of evidence, as indicated by several observational studies [13]. Another aspect under examination by numerous research groups is the biological matrix used for these tests. Most analyses, performed on fecal samples, are promising in terms of accuracy and allow for non-invasive testing by collecting small amounts of feces (~200 g). A recent meta-analysis conducted on 11 studies analyzed the diagnostic impact of PCR on fecal samples in detecting clarithromycin-resistant *H. pylori*. The test confirmed high sensitivity (91%) and specificity (96%), with an accuracy of 0.94. However, although the analysis revealed considerable heterogeneity due to numerous factors such as sample size, purification and amplification methodology used, and mutation localization, the authors suggest its use in a real-world setting [69]. In recent years, fecal samples have been widely used in the diagnosis of infectious gastroenteritis thanks to the inclusion of multiplex PCR syndromic tests in diagnostic routines. These standardized tests have revolutionized microbiological diagnostics; in addition to being economical and useful for performing a rapid differential diagnosis, they use closed systems that limit operator intervention, significantly reducing the risk of contamination. However, none of the tests currently on the market includes the search for *H. pylori* nucleic acid and resistance genes [70]. Only a recent study performed by Leonardi et al. evaluated the diagnostic accuracy of real-time PCR in stool samples compared to SAT from 100 patients with intestinal parasitosis. The molecular test showed high sensitivity (94%) and specificity (93%), demonstrating the ability to detect the presence of *H. pylori* DNA without any cross-reactivity with other intestinal pathogens. This study is promising and encourages the design of new primer targets for intestinal pathogens, including *H. pylori*, in order to implement the possibility of incorporating *H. pylori* diagnostics into a multiplex PCR panel for syndromic testing [71]. A small portion of studies have been conducted on gastric biopsies, which are collected invasively and widely used in conventional methods but less prevalent in molecular diagnostics. Furthermore, this test can be performed directly from a biopsy sample or after bacterial growth. A recent meta-analysis conducted on 6588 samples from 44 different studies highlighted the diagnostic impact of a genotypic antibiogram compared to a phenotypic antibiogram. Among the examined studies, half were conducted directly on biopsy samples, while the other half were conducted on *H. pylori* colonies from gastric biopsy. The analysis revealed that PCR performed on biopsy or colony samples also had high sensitivity (~95%) and specificity (~96%) in detecting genes associated with clarithromycin and quinolone resistance. In the first case, the best diagnostic performance was related to the combined detection of A2142G/C A2143G mutations. However, the application of molecular techniques from colonies is challenging in clinical practice as it requires the growth of a large number of bacterial colonies (dependent on the kit used). For this reason, its use is not recommended by the Maastricht IV/Florence Consensus Report [13,72]. Molecular methods are widely used in the diagnosis of viral infections with the aim of quantifying viral load in a given biological sample, allowing the clinician to evaluate therapy response during patient follow-up. These techniques are also being studied for numerous bacterial strains, including *H. pylori*. In this regard, Binmaeil et al. evaluated the performance of a quantitative PCR (qPCR) multiplex assay on 571 gastric biopsies to quantify the bacterial load present in them. All samples underwent culture examination, which yielded positive results in only 59 cases. These 59 samples underwent qPCR multiplex assay, detecting a colony quantity ranging from 10^1^ to 10^6^ CFU/mL. According to the authors, this test could not only ensure better patient follow-up, but also overcome one of the limitations associated with conventional techniques, namely the low concentration of bacteria in the biopsy sample. However, this evaluation deserves investigations in a broader context and with standardized cut-offs, which are currently unavailable [73]. A significant advantage of molecular approaches is their rapid execution, especially compared to conventional methods. This leads to a huge reduction in TAT, allowing the clinician to start antibiotic therapy early. In this context, Shan et al. analyzed the performance of a new allele-specific multiplex PCR performed on 25 gastric biopsies and compared it with real-time PCR and gene sequencing. The new method showed complete agreement with the other two techniques in evaluating the positivity rate of clarithromycin-resistant *H. pylori* (11/25; 44%). Although the application of this new technique needs further confirmation on a larger number of samples, it is of interest to note that the overall duration was only two hours from the arrival of the biological sample in the laboratory to the post-analytical phase. This significant reduction in TAT supports the crucial role of “fast microbiology” in clinical practice [74]. In line with what has been said so far, PCR performed on feces could be the most promising tool for the diagnosis of *H. pylori* infection due to its numerous strengths: (i) high sensitivity and specificity, (ii) rapid execution, (iii) low costs, (iv) significant reduction in TAT compared to conventional methods (<1 day using PCR vs. >7 days through culture isolation), and (v) possibility of performing genotypic antibiogram directly from biological samples, overcoming the issue of bacterial growth. However, molecular techniques need confirmation through conventional methods, especially regarding fastidious pathogens, because (i) bacterial DNA positivity does not imply vitality, (ii) evaluation of resistance gene expression does not necessarily imply translation into proteins, and (iii) the genotypic antibiogram does not provide a MIC value [75,76,77]. However, the execution of PCR on biopsy samples remains controversial in light of the few studies in the literature. Potential biomarkers genes which can be used for the molecular diagnosis of *H. pylori* infection are summarized in Table 3.

### 4.3. Whole Genome Sequencing

In recent years, omics technologies have emerged in a variety of fields, including microbiology. One of the main aims of this new approach is to analyze the whole genome sequencing (WGS) of target microorganisms to assess the presence of resistance mechanisms, through four different stages: (i) preparation of clones including the entire genome of the target microorganism, (ii) collection of DNA sequences of clones, (iii) generation of contig assembly, and (iv) preparation of the database [78]. Currently, the data about the use of WGS in the *H. pylori* genomic analysis are quite limited. Domanovich-Asor et al. characterized 48 Israeli *H. pylori* isolates from gastric biopsy by WGS and subsequent phylogenetic analysis. At the same time, the isolates were subjected to phenotypic antibiogram. This latter showed resistance rates for amoxicillin of 10%, for clarithromycin of 54%, for levofloxacin of 2%, for metronidazole of 31%, and for rifampicin of 4%, while 18% of the strains were MDR. WGS allowed detection of, besides the common resistance genes detectable by PCR analysis, the novel T593S variant of the *pbp1A* gene in both susceptible and resistant isolates [79]. The same research group used a dataset of 1040 genetic sequences of *H. pylori* from a worldwide dataset. The most common point mutations at pbp1A that correlated with amoxicillin resistance were S589G, S417T, and E406A (with a prevalence of 49%, 35%, and 35%, respectively). 23S rRNA A2143G, T2182C, and A2142G-C mutations, which are related to clarithromycin resistance, were found among 27%, 26%, and 6% of genomes, respectively. Mutations in levofloxacin resistance regions were present in 11–15% of cases, while mutations in the *rpoB* gene were observed in 0.3% of cases. Common mutations among the *rdxA* gene were R131K (66%), T31E (58%), and H97Y-T (22%), and those among the frxA gene were C193S (63%), F72S (59%), and G73S (58%). Overall, 93 novel variants were identified in the analysis [80]. A recent study performed in Shanghai showed the presence of mutations in the genome of *H. pylori* isolated from 112 gastric biopsy samples. A phenotypic antibiogram revealed high resistance rates for levofloxacin, metronidazole, and clarithromycin (35%, 65%, and 16%, respectively). Subsequent genomic analysis revealed the presence of the well-known intrinsic resistance mutations to the three drug classes, specifically, N87T/I and/or D91G/Y mutations in *gyrA* for levofloxacin, I38V mutation in *fdxB* for metronidazole, and A2143G and 23G for clarithromycin [81]. These studies have demonstrated how the application of WGS allows the simultaneous detection of more genes than PCR with a considerable depth of sequencing. In addition, its application allows the identification of new variants useful for epidemiological surveillance. However, there are some limitations: (i) higher costs, (ii) requirement of highly trained staff, (iii) need to evaluate a large quantity of data, (iv) necessity for a continuous update of the database to avoid a possible underestimation of the data, and (v) higher TAT (~7 days). These issues make its introduction into international guidelines and routine diagnostics difficult, but it is applicable for epidemiological studies [82]. Indeed, according to The Maastricht IV/Florence Consensus Report, WGS bears promise to allow more precise prediction of antibiotic resistance phenotypes, including those with many contributing mutations, such as metronidazole or amoxicillin resistance [13]. 

## 5. Conclusions

Table 4 summarizes the advantages and disadvantages of the different microbiological approaches to *H. pylori* detection. The pandemic of AMR can and should be countered by implementing robust and inexpensive new tools into the diagnostic routine [83]. Alongside conventional approaches, molecular techniques are increasingly being developed that can identify the presence of *H. pylori* and genes associated with antibiotic resistance in a short time. However, these methods require phenotypic confirmation, which is complicated by the particular biochemical and metabolic properties of the bacterium that make it difficult to isolate in culture. Currently, studies in the literature regarding new molecular approaches mainly include the use of stool samples, rather than gastric biopsy. In contrast, studies on culture isolation from stool samples have not been followed up, thereby not allowing the accuracy of this test to be defined. At the same time, WGS techniques are difficult to implement in the diagnostic algorithm, and thus do not find wider application in epidemiological studies to evaluate the circulation of new variant strains. A change in the diagnostic algorithm should involve the simultaneous use of conventional and molecular methods to detect the pathogen and initiate targeted therapy for the patient. Overall, the correlation between the detection of drug-resistant genes and bacterial drug-resistant phenotypes is still a critical problem in clinical practice. While advancements in molecular biology have enabled the identification of specific genetic markers associated with AMR in bacteria, the translation of this genetic information into accurate predictions of bacterial drug resistance phenotypes is complex and often imperfect [84]. One of the primary issues lies in the multifactorial nature of AMR. Consequently, the presence of drug-resistant genes does not always directly translate into observable drug-resistant phenotypes. Moreover, the interaction between genetic factors and environmental conditions further complicates the prediction of bacterial drug resistance. Factors such as the local prevalence of resistance genes, antibiotic usage patterns, and microbial population dynamics can influence the expression and dissemination of drug resistance within bacterial communities [85]. In addition, the rapid evolution of bacterial pathogens poses an ongoing challenge in keeping up with the emergence of new resistance mechanisms. As bacteria evolve and adapt in response to selective pressures, the efficacy of existing diagnostic methods and treatment strategies may diminish over time. Addressing this challenge requires a multifaceted approach that integrates molecular analyses with phenotypic assays and clinical data [86]. Additionally, concerted efforts are needed to implement robust surveillance systems for monitoring the prevalence and spread of drug-resistant bacteria in clinical settings. Combining genomic surveillance with epidemiological data can improve patient management. At the same time, there is a need for new molecules against MDR pathogens [87,88]. Recently, the peptide nucleic acid–fluorescence in situ hybridization (PNA-FISH) technique has emerged as a novel approach. This method, applicable to histological samples, is characterized by high sensitivity and specificity (97% and 100%, respectively) in diagnosing *H. pylori* infection. It enables the identification of pathogens that are often undetectable with standard histological examination. Moreover, PNA-FISH stands out because of its speed, accuracy, and cost-effectiveness in detecting clarithromycin resistance in *H. pylori* strains from gastric biopsies. However, despite its benefits in simultaneously detecting the bacterium and its clarithromycin resistance, the disadvantages of PNA-FISH, such as its laborious preparation, and the need for a fluorescent microscope and specific expertise for result interpretation, may limit its use [89]. New real-life studies are urgently required in order to better define the new changes in the diagnostic algorithm of *H. pylori* infection in the treatment-failure era.

## Figures and Tables

**Figure 1 antibiotics-13-00357-f001:**
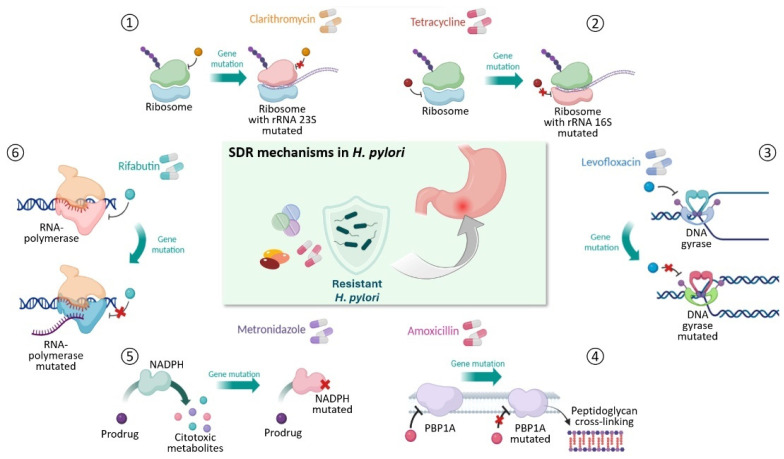
Main SDR mechanisms observed in *H. pylori*.

**Figure 2 antibiotics-13-00357-f002:**
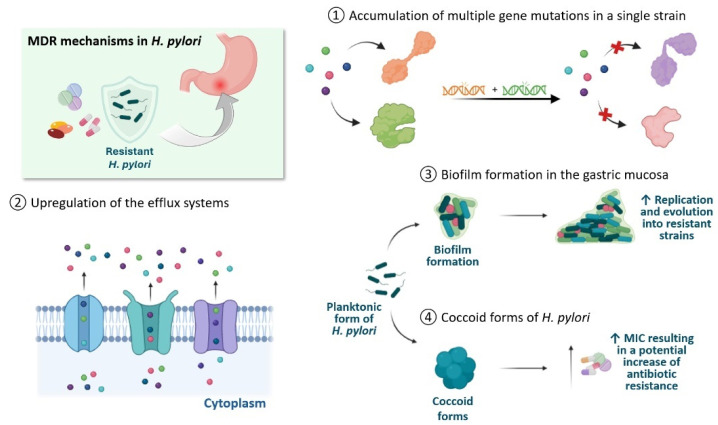
Main MDR mechanisms observed in *H. pylori*.

**Table 1 antibiotics-13-00357-t001:** Sensitivity and specificity of non-invasive method for diagnosing *H. pylori* infection.

Non-Invasive Methods	Sensibility	Specificity
^13^C-UBT	96.60%	96.93%
^14^C-UBT	96.15%	89.84%
SAT	95.5%	97.6%
Serological test	80–95%	80–95%

Abbreviations: UBT, urea breath test; SAT; stool antigen test.

**Table 2 antibiotics-13-00357-t002:** Sensitivity, specificity, and accuracy of each endoscopic technique for diagnosing *H. pylori* infection.

Endoscopic Techniques	Sensibility	Specificity	Accuracy
WLI	90.00%	70.00%	78.00%
NBI	85.00%	80.00%	82.00%
LCI	95.00%	76.70%	84.00%
BLI	95.00%	80.00%	86.00%

Abbreviations: WLI, white light endoscopy; NBI; narrow-band imaging; LCI, linked color imaging; BLI, blue laser imaging.

**Table 3 antibiotics-13-00357-t003:** Summary of the different potential biomarker genes for the molecular diagnosis of *H. pylori* infection.

Gene Target	Applicability
*cagA*, *vacA*, *ureA*, *ureC*	Identification
A2143G, A2142G, A2142C	Detection of clarithromycin resistance
*gyrA*, *gyrB*	Detection of levofloxacin resistance
*pbp1A*, *pbp2*, *pbp3*, *hefC*, *hopC*, *hofH*	Detection of amoxicillin re-sistance
*TET-1*	Detection of tetracycline re-sistance

**Table 4 antibiotics-13-00357-t004:** Advantages and disadvantages of the different microbiological approaches about the *H. pylori* detection.

Microbiological Approach	Advantages	Disadvantages
Culture isolation and phenotypic antibiogram	Diagnostic gold standard Defines a MIC Falls within the Maastricht IV/Florence Consensus Report	Difficult to perform Higher TAT (>7 days) Low sensitivity
PCR and genotypic antibiogram	High sensitivity and specificity Performed directly on biological sample Significant reduction in TAT (<1 day) Promotes the differential diagnosis with other gastro-intestinal tract infection Falls within the Maastricht IV/Florence Consensus Report	Needs of a confirmation through conventional microbiological approach Limited resistance gene detection Does not define a MIC
WGS	Simultaneous detection of more genes with an elevated depth of sequencing Useful for the identification of new variants and epidemiological surveillance	Higher costs Requirement of highly trained staff Need to evaluate a large quantity of data Necessity to continuously update the database to avoid a possible underestimation of the data Higher TAT (~7 days) Not included within the Maastricht IV/Florence Consensus Report

Abbreviations: MIC, minimal inhibitory concentration; TAT, turn-around time; PCR, polymerase chain reaction; WGS, whole genome sequencing.

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
