# Peer review of "Change in Diagnosis of *Helicobacter pylori* Infection in the Treatment-Failure Era"

_antibiotics, 2024, doi:10.3390/antibiotics13040357_

Round 1
Reviewer 1 Report
Comments and Suggestions for Authors
The author reviews the current diagnostic methods for HP. These directions are mature and have been used in clinical practice. There have been many review articles on these diagnostic techniques. I don't think the author's focus is right.
1. The main reason for the failure of HP treatment is drug resistance. The authors can add to the review of mechanisms of bacterial resistance generation. This adds to the novelty of the manuscript.
2. At present, the culture of HP is not used clinically, mainly because of the difficulty of culture, slow growth, and low feasibility of further drug resistance test. Therefore, bacterial drug resistance gene detection technology is the current development trend. However, the correlation between the detection of drug-resistant genes and bacterial drug-resistant phenotypes is still an inconclusive problem in clinical practice, which invites the author to add relevant discussion content.
In general, the content of the review is outdated and does not pay enough attention to cutting-edge developments.
Comments on the Quality of English LanguageLanguage quality is acceptable
Author Response
The author reviews the current diagnostic methods for HP. These directions are mature and have been used in clinical practice. There have been many review articles on these diagnostic techniques. I don't think the author's focus is right.
1. The main reason for the failure of HP treatment is drug resistance. The authors can addto the review of mechanisms of bacterial resistance generation. This adds to the novelty of the manuscript.
Reply 1. Thank you for your comment. Section 2 focuses on discussing the mechanisms of H. pylori resistance. However, thanks to your suggestion, we have expanded the section with other important information (see lines 117-123, 129-133, 154-156, 172-174, 188-190). Moreover, as suggested by Reviewer #4, we considered the possible resistance pattern of aminoglycosides, for which, however, there is little evidence in the literature specific to H. pylori infection (see lines 224-229). In addition, Table 3 highlights the application of the resistance genes discussed in clinical practice (see lines 538-539).
2. At present, the culture of HP is not used clinically, mainly because of the difficulty of culture, slow growth, and low feasibility of further drug resistance test. Therefore, bacterial drug resistance gene detection technology is the current development trend. However, the correlation between the detection of drug-resistant genes and bacterial drug-resistant phenotypes is still an inconclusive problem in clinical practice, which invites the author to add relevant discussion content.
Reply 2. Thank you for your comment. The manuscript has been revised (see lines 601-620).
In general, the content of the review is outdated and does not pay enough attention to cutting-edge developments
Reviewer 2 Report
Comments and Suggestions for Authors
• What is the main question addressed by the research?
The main question addressed by the research is H. pylori drug resistance and treatment failure
• Do you consider the topic original or relevant in the field? Does it address a specific gap in the field? Please also explain why this is/ is not the case.
Honestly, it doesn't address the gap in the field but this field needs this review to read and get an idea, especially for new starters.
• What does it add to the subject area compared with other published material?
It shows the emergency of new drug target research and the development of new methods for overcoming treatment failures
• What specific improvements should the authors consider regarding the methodology? What further controls should be considered?
The authors do not need to consider any specific improvements to the methodology.
• Are the conclusions consistent with the evidence and arguments presented and do they address the main question posed? Please also explain why this is/ is not the case.
Yes, all of them enough to address the main question
• Are the references appropriate?
The references are appropriate (Between 2023-2018)
• Please include any additional comments on the tables and figures.
Figures are suitable for the topic and all referenced.
Author Response
- What is the main question addressed by the research?
The main question addressed by the research is H. pylori drug resistance and treatment failure
- Do you consider the topic original or relevant in the field? Does it address a specific gap in the field? Please also explain why this is/ is not the case. Honestly, it doesn't address the gap in the field but this field needs this review to read and get an idea, especially for new starters.
- What does it add to the subject area compared with other published material? It shows the emergency of new drug target research and the development of new methods for overcoming treatment failures
- What specific improvements should the authors consider regarding the methodology? What further controls should be considered? The authors do not need to consider any specific improvements to the methodology.
- Are the conclusions consistent with the evidence and arguments presented and do they address the main question posed? Please also explain why this is/ is not the case. Yes, all of them enough to address the main question
- Are the references appropriate? The references are appropriate (Between 2023-2018)
- Please include any additional comments on the tables and figures. Figures are suitable for the topic and all referenced.
Reply: Thank you for your positive comments.
Reviewer 3 Report
Comments and Suggestions for Authors
Review report:
The authors focused on this review mainly the molecular mechanisms of antimicrobial single drug resistance (SDR) and multi-drug resistance (MDR). Detailed explanation of non-invasive and invasive methods, each characterized by their advantages, disadvantages, and limitations. Finally, they discussed new perspectives in diagnostics approaches (Diagnostics, microbiological, whole genome sequencing etc.,) to better define the new changes in the diagnostic algorithm of H. pylori infection in the treatment failure era.
Altogether this review is informative and well written.
Suggestions:
1.The strength of this review is mainly the author's clearly written step by step especially SDR and MDR molecular mechanisms observed in H. pylori using Diagrammatic approach. Fig 1 resolution is less especially labelling (If possible, get better resolution)
2.If possible, please include the table it contains sensitivity and specificity of various diagnostic approaches (advantages and disadvantages) in subsection of Conventional diagnostic approaches. (UBT, SAT, serological etc.,)
Author Response
The authors focused on this review mainly the molecular mechanisms of antimicrobial single drug resistance (SDR) and multi-drug resistance (MDR). Detailed explanation of non-invasive and invasive methods, each characterized by their advantages, disadvantages, and limitations. Finally, they discussed new perspectives in diagnostics approaches (Diagnostics, microbiological, whole genome sequencing etc.,) to better define the new changes in the diagnostic algorithm of H. pylori infection in the treatment failure era.
Altogether this review is informative and well written.
Suggestions:
1.The strength of this review is mainly the author's clearly written step by step especially SDR and MDR molecular mechanisms observed in H. pylori using Diagrammatic approach. Fig 1 resolution is less especially labelling (If possible, get better resolution)
Reply 1. Thank you for your comment. Figure 1 has been revised (see line 138).
2.If possible, please include the table it contains sensitivity and specificity of various diagnostic approaches (advantages and disadvantages) in subsection of Conventional diagnostic approaches. (UBT, SAT, serological etc.,).
Reply 2. Thank you for your comment. Table 1 have been added (see lines 351-352).
Reviewer 4 Report
Comments and Suggestions for Authors
1. AMR with respect to H. pylori is included, however general statistics and present scenario about AMR can be included in the introduction to have deeper impact of AMR.
2. Text in Fig.2 is not clear. Better resolution might be needed to read the text clearly.
3. Aminoglycosides were being used for the treatment of H. pylori infection. Is there aminoglycoside modifying enzyme mediated resistance reported on H. pylori, can be included if any?
4. Can include PNA-FISH tests and its limitations. Doi: 10.3748/wjg.v21.i40.11221
5. Line: 453, H. pylori italics.
6. If possible molecular probes or the potential biomarkers genes which can be used for the molecular level detection/diagnosis can be tabulated for easy reader understand and for future scope to work on it.
Author Response
- AMR with respect to H. pylori is included, however general statistics and present scenario about AMR can be included in the introduction to have deeper impact of AMR.
Reply 1. Thank you for your comment. The manuscript has been revised (see lines 84-91).
- Text in Fig.2 is not clear. Better resolution might be needed to read the text clearly.
Reply 2. Thank you for your comment. Figure 2 has been revised (see line 255).
- Aminoglycosides were being used for the treatment of H. pylori infection. Is there aminoglycoside modifying enzyme mediated resistance reported on H. pylori, can be included if any?
Reply 3. Thank you for your comment. The manuscript has been revised (see lines 224-229).
- Can include PNA-FISH tests and its limitations. Doi: 10.3748/wjg.v21.i40.11221.
Reply 4. Thank you for your comment. The manuscript has been revised (see lines 621-631).
- Line: 453, H. pylori italics.
Reply 5. Thank you for your comment. The manuscript has been revised (see line 485).
- If possible molecular probes or the potential biomarkers genes which can be used for the molecular level detection/diagnosis can be tabulate d for easy reader understand and for future scope to work on it.
Reply 6. Thank you for your comment. Table 3 have been added (see lines 538-539).
Round 2
Reviewer 1 Report
Comments and Suggestions for Authors
I accept the revisions by authors and agree to publish this paper
The manuscript can be read in English